# Hidden Markov Neural Networks

**DOI:** 10.3390/e27020168

**Published:** 2025-02-05

**Authors:** Lorenzo Rimella, Nick Whiteley

**Affiliations:** 1Dipartimento di Scienze Economico-Sociali e Matematico-Statistiche, University of Torino, 10124 Torino, Italy; 2Statistics Initiative, Collegio Carlo Alberto, 10124 Torino, Italy; 3School of Mathematics, University of Bristol, Fry Building, Woodland Road, Bristol BS8 1UG, UK; nick.whiteley@bristol.ac.uk

**Keywords:** Bayesian neural networks, Hidden Markov models, variational inference

## Abstract

We define an evolving in-time Bayesian neural network called a Hidden Markov Neural Network, which addresses the crucial challenge in time-series forecasting and continual learning: striking a balance between adapting to new data and appropriately forgetting outdated information. This is achieved by modelling the weights of a neural network as the hidden states of a Hidden Markov model, with the observed process defined by the available data. A filtering algorithm is employed to learn a variational approximation of the evolving-in-time posterior distribution over the weights. By leveraging a sequential variant of Bayes by Backprop, enriched with a stronger regularization technique called variational DropConnect, Hidden Markov Neural Networks achieve robust regularization and scalable inference. Experiments on MNIST, dynamic classification tasks, and next-frame forecasting in videos demonstrate that Hidden Markov Neural Networks provide strong predictive performance while enabling effective uncertainty quantification.

## 1. Introduction

Hidden Markov models (HMMs) are an efficient statistical tool for identifying patterns in dynamic datasets, with applications ranging from speech recognition [1] to computational biology [2]. Neural networks (NNs) are currently the most popular models in machine learning and artificial intelligence, demonstrating outstanding performance across several fields. In this paper, we propose a novel hybrid model called a Hidden Markov Neural Network (HMNN), which combines Factorial Hidden Markov models [3] and neural networks.

Intuitively, we aim to perform Bayesian inference on a time-evolving NN. However, computing the posterior distribution over the weights of even a static NN is a complex and generally intractable task. The extensive literature on variational Bayes [4] and its success in Bayesian inference for NNs [5,6,7] has motivated us to adopt this technique in HMNNs. In particular, the resulting procedure becomes the sequential counterpart of the Bayes by the Backprop algorithm proposed by [7]. As in [7], the reparameterization trick [6] plays a pivotal role in generating unbiased and low-variance estimates of the gradient.

HMNNs are particularly suited for time-series forecasting and continual learning [8]. As noted by [9], much of the research in this area has focused on preventing forgetting [10,11,12]. However, sudden changes in the statistical properties of the data may be an intrinsic feature of the generating process. In such cases, preserving all prior knowledge is not desirable, and it becomes necessary to forget irrelevant information. This can be achieved through the application of a Markov transition kernel [9], which can be interpreted as the transition kernel of an HMM. In this sense, HMNNs adopt the adaptation idea proposed by [9] and extend it by leveraging the well-established framework of HMMs, further generalizing it to a broader class of variational approximations and stochastic kernels.

An alternative to variational Bayes is particle approximation via Sequential Monte Carlo [13]. However, these algorithms are known to suffer from the curse of dimensionality [14], making them unfeasible even for simple NN architectures.

## 2. Hidden Markov Neural Networks

For a time horizon T∈N, a Hidden Markov model (HMM) is a bivariate process composed of an unobserved Markov chain ${\left(W_{t}\right)}_{t=0,\ldots ,T}$, called the hidden process, and a collection of observations ${\left(D_{t}\right)}_{t=1,\ldots ,T}$, called the observed process, where the single observation at *t* is conditionally independent of all the other variables given the hidden process at the corresponding time step. We consider the case where the latent state-space is $R^{V}$, with *V* finite set, and $D_{t}$ is valued in D whose form is model-specific (discrete, $R^{d}$ with d∈N, etc.). To describe the evolution of an HMM, three quantities are needed: the initial distribution, the transition kernel and the emission distribution. We use $\lambda_{0}\left(\cdot \right)$ for the probability density function of $W_{0}$ (initial distribution). We write $p(W_{t-1},\cdot )$ for the conditional probability density function of $W_{t}$ given $W_{t-1}$ (transition kernel of the Markov chain). We call $g(W_{t},D_{t})$ the conditional probability mass or density function of $D_{t}$ given $W_{t}$ (emission distribution). We remark that HMM is often used for finite state spaces, while state-space model is used for continuous state spaces; here, following [14], we consider the two terminologies exchangeable.

We introduce a novel HMM called a Hidden Markov Neural Network (HMNN) where the hidden process describes the evolution of the weights of a neural network. For instance, in the case of feed-forward neural networks, the finite set *V* collects the location of each weight and v∈V can be thought of as a triplet (l,i,j), indicating that the weight $W_{t}^{v}$ is a weight of the NN at time *t*, and is precisely related to the connection of the hidden unit *i* (or input feature *i* if l−1=0) in the layer l−1 with the hidden unit *j* in the layer *l* (which might be the output layer). In such a model, we also assume that the weights evolve independently from each other, meaning that the transition kernel factorizes as follows:(1)$$p\left(W_{t-1},W_{t}\right)=\prod_{v\in V}p^{v}\left(W_{t-1}^{v},W_{t}^{v}\right),\;W_{t-1},W_{t}\in R^{V}.$$

Under this assumption, an HMNN is a well-known class of HMM called a Factorial Hidden Markov model (FHMM) [3,15,16]; see Figure 1 for a graphical representation. Note that this factorization is key to ensuring that the computational cost of performing inference does not exponentially increase, and, in some scenarios, it allows us to perform close-form calculations; see Section 2.3. As an alternative, one could use a block structure [14], which requires, however, for the user to keep track of the correlations within blocks.

There is no restriction on the form of the neural network or the data $D_{t}$; however, we focus on a feed-forward neural network framework and on a supervised learning scenario where the observed process is composed by an input $x_{t}$ and output $y_{t}$, such that the neural network associated to the weights $W_{t}$ maps the input onto a probability distribution on the space of the output, which represents the emission distribution $g(W_{t},D_{t})$.

### 2.1. Filtering Algorithm for HMNN

The filtering problem aims to compute the conditional distributions of $W_{t}$ given $D_{1},\ldots ,D_{t}$, which is called filtering distribution. In this paper, we use the operator notation from [14], which gives us a way to compactly represent the filtering algorithm. We refer to [13,14] for a thorough review.

We denote the filtering distribution with $\pi_{t}$ and, ideally, we can compute it with a forward step through the data: (2)$$\pi_{0}:=\lambda_{0},\;\pi_{t}:=C_{t}P\pi_{t-1},$$
where P and $C_{t}$ are called the prediction operator and correction operator, respectively, and they are defined as follows: (3)$$P\pi_{t-1}\left(A\right):=\int I_{A}\left(W_{t}\right)p\left(W_{t-1},W_{t}\right)\pi_{t-1}\left(W_{t-1}\right)dW_{t-1}dW_{t},$$(4)$$C_{t}\pi_{t}\left(A\right):=\frac{\int I_{A}\left(W_{t}\right)g\left(W_{t},D_{t}\right)\pi_{t}\left(W_{t}\right)dW_{t}}{\int g\left(W_{t},D_{t}\right)\pi_{t}\left(W_{t}\right)dW_{t}},$$
with $\pi_{t-1},\pi_{t}$ as the probability density functions, $I_{\cdot }\left(\cdot \right)$ as the indicator function, and *A* as the set in the sigma field of $R^{V}$. Throughout this paper, we refer to our target distribution as the filtering distribution $\pi_{t}$, which is indeed the time-evolving posterior distribution over the weights of our NN. Recursion (Equation 2) is intractable for any non-linear architecture of the underlying neural network. As a solution, we can apply variational inference, which allows us to approximate a posterior distribution when operations cannot be performed in closed form. The use of variational inference is then crucial to allow us to use any activation function in our NN.

Variational inference can be used to sequentially approximate the target distribution $\pi_{t}$ with a variational approximation $q_{\theta_{t}}$ belonging to a pre-specified class of distributions Q. The approximate distribution $q_{\theta_{t}}$ is uniquely identified inside the class of distributions by a vector of parameters $\theta_{t}$, which is chosen to minimize a Kullback–Leibler (KL) divergence criteria: (5)$$q_{\theta_{t}}:=\arg\min_{q_{\theta }\in Q}\mathrm{KL}(q_{\theta }\left|\right|\pi_{t})=\arg\min_{q_{\theta }\in Q}\mathrm{KL}(q_{\theta }\left|\right|C_{t}P\pi_{t-1}),$$
where the Kullback–Leibler divergence can be rewritten as(6)$$\mathrm{KL}(q_{\theta }||C_{t}P\pi_{t-1})=const.+\mathrm{KL}(q_{\theta }||P\pi_{t-1})-E_{q_{\theta }\left(w\right)}\left[\log\left(g(w,D_{t})\right)\right],$$
because of the properties of the correction operator (Equation 4). From the above representation, we can also observe that minimizing the Kullback–Leibler divergence is equivalent to maximizing the evidence lower bound (ELBO):(7)$$\mathrm{ELBO}(q_{\theta };\pi_{t-1},D_{t})=E_{q_{\theta }\left(w\right)}\left[\log\left(g(w,D_{t})\right)\right]-\mathrm{KL}(q_{\theta }||P\pi_{t-1}).$$
Sequential training using (Equation 5) is again intractable because it requires the filtering distribution at time t−1, i.e., $\pi_{t-1}$. Although under good optimization, we could consider $q_{\theta_{t}}\approx \pi_{t}$ when $\pi_{t-1}$ is known, similarly, $q_{\theta_{t-1}}\approx \pi_{t-1}$ when $\pi_{t-2}$ is known, and so on. Given that $\pi_{0}$ is our prior knowledge $\lambda_{0}$ on the weights before training, we can find a $q_{\theta_{1}}$ that approximates $\pi_{1}$ using (Equation 5), and then we can propagate forward our approximation by following the previous logic. In this way, an HMNN is trained sequentially on the same flavour of (Equation 5), by substituting the optimal filtering with the last variational approximation. As for the optimal procedure, we define an approximated filtering recursion, where ${\tilde{\pi }}_{t}$ stands for the sequential variational approximation of $\pi_{t}$: (8)$${\tilde{\pi }}_{0}:=\lambda_{0},\;{\tilde{\pi }}_{t}:=V_{Q}C_{t}P{\tilde{\pi }}_{t-1},$$
where the operators $P,C_{t}$ are as in recursion (Equation 2) and the operator $V_{Q}$ is defined as follows: (9)$$V_{Q}\rho :=\arg\min_{q_{\theta }\in Q}\mathrm{KL}(q_{\theta }||\rho ),$$
with ρ being a probability distribution and Q being the class of variational distributions, e.g., in (Equation 8), we have $\rho =C_{t}P{\tilde{\pi }}_{t-1}$. Note that, in this final notation, we are hiding the dependence on the variational parameters θ.

Observe that this approach follows the assumed density filter paradigm [17,18], where the true posterior is projected onto a family of distributions which are easy to work with, and then propagated forward.

### 2.2. Sequential Reparameterization Trick

The minimization procedure exploited in recursion (Equation 8) cannot be solved in a closed form and we propose to find a suboptimal solution through gradient descent. Consider a general time step *t*, where we want to approximate $\pi_{t}$ with ${\tilde{\pi }}_{t}=q_{\theta_{t}}$. This requires an estimate of the gradient of $\mathrm{KL}(q_{\theta }||C_{t}P{\tilde{\pi }}_{t-1})$. As explained in [7], if  $W∼q_{\theta }$ can be rewritten as ϵ∼ν through a deterministic transformation *h*, i.e., W=h(θ,ϵ), then(10)$$\begin{array}{cc} \frac{\partial \mathrm{KL}(q_{\theta }||C_{t}P{\tilde{\pi }}_{t-1})}{\partial \theta }= & E_{\nu (\epsilon )}{\left[\frac{\partial \log\left(q_{\theta }\left(W\right)\right)}{\partial W}\frac{\partial W}{\partial \theta }+\frac{\partial \log\left(q_{\theta }\left(W\right)\right)}{\partial \theta }\right]}_{W=h(\theta ,\epsilon )}\\ \; & \;-E_{\nu (\epsilon )}{\left[\frac{\partial \log\left(P{\tilde{\pi }}_{t-1}\left(W\right)\right)}{\partial W}\frac{\partial W}{\partial \theta }\right]}_{W=h(\theta ,\epsilon )}\\ \; & \;-E_{\nu (\epsilon )}{\left[\frac{\partial \log\left(g(W,D_{t})\right)}{\partial W}\frac{\partial W}{\partial \theta }\right]}_{W=h(\theta ,\epsilon )} \end{array}$$
where we used (Equation 6) to simplify the form of the equation (see Appendix A). Given (Equation 10), we can estimate the expectation $E_{\nu (\epsilon )}$ via straightforward Monte Carlo sampling:(11)$$\begin{array}{cc} \frac{\partial \mathrm{KL}(q_{\theta }||C_{t}P{\tilde{\pi }}_{t-1})}{\partial \theta }\approx & \frac{1}{N}\sum_{i=1}^{N}\left\{{\left[\frac{\partial \log\left(q_{\theta }\left(W\right)\right)}{\partial W}\frac{\partial W}{\partial \theta }+\frac{\partial \log\left(q_{\theta }\left(W\right)\right)}{\partial \theta }\right]}_{W=h(\theta ,\epsilon )}\right.\\ \; & \;\;\;\;-\left.{\left[\frac{\partial \log\left(P{\tilde{\pi }}_{t-1}\left(W\right)\right)}{\partial W}\frac{\partial W}{\partial \theta }\right]}_{W=h(\theta ,\epsilon )}\right.\\ \; & \;\;\;\;-{\left.{\left[\frac{\partial \log\left(g(W,D_{t})\right)}{\partial W}\frac{\partial W}{\partial \theta }\right]}_{W=h(\theta ,\epsilon )}\right\}}_{\epsilon =\epsilon^{\left(i\right)}} \end{array}$$
with $\epsilon^{\left(i\right)}∼\nu$ and *N* the size of the Monte Carlo sample. Given the Monte Carlo estimate of the gradient, we can then update the parameters $\theta_{t}$, related to the variational approximation at time *t*, according to any gradient descent technique. Algorithm 1 displays this procedure and, for the sake of simplicity, we write the algorithm with an update that follows a vanilla gradient descent.
**Algorithm 1** Approximate filtering recursionSet: ${\tilde{\pi }}_{0}=\lambda_{0}$**for** 
t=1,…,T 
**do**    Initialize: $\theta_{t}$    **repeat**       $\epsilon^{\left(i\right)}∼\nu ,$  i=1,…,N       Estimate the gradient ∇ with (Equation 11) evaluated in $\theta =\theta_{t}$       Update the parameters: $\theta_{t}=\theta_{t}-l\nabla$   **until** Maximum number of iterations   Set: ${\tilde{\pi }}_{t}=q_{\theta_{t}}$**end for****Return:** 
${\left(\theta_{t}\right)}_{t=1,\ldots ,T}$

As suggested in the literature [5,7], the cost function in (Equation 6) is suitable for minibatch optimization. This might be useful when, at each time step, $D_{t}$ is made of multiple data, and so, a full computation of the gradient is computationally prohibitive.

### 2.3. Gaussian Case

A fully Gaussian model, i.e., when both the transition kernel and the variational approximation are Gaussian distributions, is not only convenient because the form of h(θ,ϵ) is trivial, but also because there exists a closed-form solution for $P{\tilde{\pi }}_{t-1}$. Another appealing aspect of the Gaussian choice is that similar results hold for the scale mixture of Gaussians, which allows us to use a more complex variational approximation and a transition kernel of the same form as the prior distribution in [7]. We start by considering the variational approximation. We choose $q_{\theta }:={⨂}_{v\in V}q_{\theta }^{v}$, where $q_{\theta }^{v}$ is a mixture of Gaussian with parameters $\theta^{v}=\left(m^{v},s^{v}\right)$ and $\gamma^{v}$ hyperparameter. Precisely, for a given weight $W^{v}$ of the feed-forward neural network,(12)$$q_{\theta }^{v}(W^{v}):=\gamma^{v}N\left(W^{v}|m^{v},{\left(s^{v}\right)}^{2}\right)+\left(1-\gamma^{v}\right)N\left(W^{v}|0,{\left(s^{v}\right)}^{2}\right),$$
where $\gamma^{v}\in \left(0,1\right]$, $m^{v}\in R$, ${\left(s^{v}\right)}^{2}\in R_{+}$, and $N\left(\cdot |\mu ,\sigma^{2}\right)$ is the Gaussian density with mean μ and variance $\sigma^{2}$. We refer to this technique as variational DropConnect because it can be interpreted as setting the weight in position *v* of the neural network around zero with probability $1-\gamma^{v}$, and so, it plays a role of regularization similar to [19]. Under variational DropConnect, the deterministic transformation h(θ,ϵ) is still straightforward. Indeed, given that $q_{\theta }$ factorizes, then $h\left(\theta ,\epsilon \right)={\left(h^{v}\left(\theta^{v},\epsilon^{v}\right)\right)}_{v\in V}$ (each $W^{v}$ depends only on $\theta^{v}$) and $W^{v}$ is distributed as (Equation 12), which is equivalent to consider(13)$$W^{v}=\eta^{v}m^{v}+\xi^{v}s^{v},\;\mathrm{with}\;\eta^{v}∼\mathrm{Be}\left(\cdot |\gamma^{v}\right),\;\xi^{v}∼N\left(\cdot |0,1\right),$$
where Be(·|p) is the Bernoulli density with parameter *p*. Observe that the distribution of (Equation 13) is (Equation 12), as $\gamma^{v}$ comes from the Bernoulli random variable $\eta^{v}$, which activates or not the mean $m^{v}$, while the $\xi^{v}$ represents the Gaussian term. Hence, $h^{v}\left(\theta^{v},\epsilon^{v}\right)=\eta^{v}m^{v}+\xi^{v}s^{v}$, where $\theta^{v}=\left(m^{v},s^{v}\right)$ and $\epsilon^{v}=\left(\eta^{v},\xi^{v}\right)$ with $\eta^{v}$ Bernoulli with parameter $\gamma^{v}$ and $\xi^{v}$ standard Gaussian, meaning that we just need to sample from a Bernoulli and a Gaussian distribution independently. The collection of hyperparameters ${\left(\gamma^{v}\right)}_{v\in V}$ represents the variational DropConnect rate per each weight in the NN and we generally choose $\gamma^{v}=\gamma \in \left(0,1\right]$ per each v∈V, i.e., we have a global regularization parameter. Note that ${\left(\gamma^{v}\right)}_{v\in V}$ must be considered as fixed and cannot be learned during training, because from (Equation 10), we need the distribution of ϵ to not be dependent on the learnable parameters. Consider now the transition kernel. It is chosen to be a scale mixture of Gaussians with parameters ϕ,α,σ,c,μ: (14)$$\begin{array}{cc} p(W_{t-1},W_{t}) & :=\phi N\left(W_{t}|\mu +\alpha \left(W_{t-1}-\mu \right),\sigma^{2}I_{V}\right)\\ \; & \;+\left(1-\phi \right)N\left(W_{t}|\mu +\alpha \left(W_{t-1}-\mu \right),\left(\sigma^{2}c^{2}\right)I_{V}\right), \end{array}$$
where ϕ∈[0,1], $\mu \in R^{V}$, α∈[0,1), $\sigma \in R_{+}$, $I_{V}$ is the identity matrix on $R^{V,V}$, $c\in R_{+}$ and c>1. Intuitively, the transition kernel tells us how we are expecting the weights to be in the next time step given the states of the weights at the current time. We can interpret it, along with the previous variational approximation, as playing the role of an evolving prior distribution which constrains the new posterior distribution in regions that are determined from the previous training step. The choice of the transition kernel is crucial. A too conservative kernel would constrain too much training and the algorithm would not be able to learn patterns in new data. On the contrary, a too flexible kernel could just forget what was learned previously and adapt to the new data only.

As we are considering Gaussian distribution, we can solve $P{\tilde{\pi }}_{t-1}$ in closed form, and, precisely, we obtain another scaled mixture of Gaussian distributions. We can then directly work on the Gaussian density $P{\tilde{\pi }}_{t-1}\left(W_{t-1}\right)$, which is a product over v∈V of scaled mixture of Gaussian densities. Specifically, consider a general weight v∈V and call ${\left(P{\tilde{\pi }}_{t-1}\right)}^{v}\left(W_{t-1}^{v}\right)$ the marginal density of $P{\tilde{\pi }}_{t-1}$ on the component *v*. If $m_{t-1}^{v},s_{t-1}^{v}$ are the estimates of $m^{v},s^{v}$ at time t−1, then(15)$$\begin{array}{cc} {\left(P{\tilde{\pi }}_{t-1}\right)}^{v}\left(W_{t-1}^{v}\right)= & \gamma^{v}\phi N\left(W_{t-1}^{v}|\mu^{v}-\alpha \left(\mu^{v}-m_{t-1}^{v}\right),\sigma^{2}+\alpha^{2}{\left(s_{t-1}^{v}\right)}^{2}\right)\\ \; & +\left(1-\gamma^{v}\right)\phi N\left(W_{t-1}^{v}|\mu^{v}-\alpha \mu^{v},\sigma^{2}+\alpha^{2}{\left(s_{t-1}^{v}\right)}^{2}\right)\\ \; & +\gamma^{v}\left(1-\phi \right)N\left(W_{t-1}^{v}|\mu^{v}-\alpha \left(\mu^{v}-m_{t-1}^{v}\right),\sigma^{2}/c^{2}+\alpha^{2}{\left(s_{t-1}^{v}\right)}^{2}\right)\\ \; & +\left(1-\gamma^{v}\right)\left(1-\phi \right)N\left(W_{t-1}^{v}|\mu^{v}-\alpha \mu^{v},\sigma^{2}/c^{2}+\alpha^{2}{\left(s_{t-1}^{v}\right)}^{2}\right). \end{array}$$
We can observe that (Equation 15) is again a scale mixture of Gaussians, where all the variances are influenced by the variances at the previous time step according to $\alpha^{2}$. On the one hand, the variational DropConnect rate $\gamma^{v}$ tells how to scale the mean of the Gaussians according to the previous estimates $m_{t-1}^{v}$. On the other hand, ϕ controls the entity of the jumps by allowing the weights to stay in place with a small variance $\sigma^{2}/c^{2}$ and permitting big jumps with $\sigma^{2}$ if necessary. As in [7], when learning $s_{t}$, we use the transformation ${\tilde{s}}_{t}$ such that $s_{t}=\log\left(1+\exp\left({\tilde{s}}_{t}\right)\right)$.

Across Section 2, we have declared multiple quantities which we summarize in Table 1.

### 2.4. Performance and Uncertainty Quantification

After running Algorithm 1, we obtain a sequence of variational approximations ${\tilde{\pi }}_{t}=q_{\theta_{t}}$ for our filtering distributions $\pi_{t}$. These approximations represent the posterior distribution of $W_{t}$, meaning that we are approximating the distribution of $W_{t}|D_{1},\ldots ,D_{t}$ with a known distribution. Given a realization of $W_{t}$, we can compute $g(W_{t},\cdot )$, which represents the likelihood at time *t* or, equivalently, the performance of the neural network (NN) with weights $W_{t}$. We generally assume the ability to compute the mean of ${\tilde{\pi }}_{t}=q_{\theta_{t}}$, i.e., the posterior mean, and to sample from it, i.e., obtain posterior samples. These two components allow us to assess the performance of our HMNN over time and evaluate how certain we are about that performance.

In our experiments, we typically refer to performance when evaluating our time-evolving NN using the posterior mean, $E_{{\tilde{\pi }}_{t}\left(W_{t}\right)}\left[W_{t}\right]$. Conversely, when assessing uncertainty, we consider posterior samples, $W_{t}^{\left(1\right)},\ldots ,W_{t}^{\left(N\right)}$.

## 3. Related Work

### 3.1. Combining NNs and HMMs

Multiple attempts have been made in the literature to combine HMMs and NNs. In [20], an NN is trained to approximate the emission distribution of an HMM. Refs. [21,22] preprocess the data with an NN and then use the output as the observed process of a discrete HMM. Ref. [23] proposes “hidden neural networks” where NNs are used to parameterize Class HMM, an HMM with a distribution over classes assigned to each state. Other recent works include [24,25,26]. In neuroscience, Ref. [27] explores the idea of updating measures of uncertainty over the weights in a mathematical model of a neuronal network as part of a “Bayesian Plasticity” hypothesis of how synapses take uncertainty into account during learning. However, they did not focus on artificial neural networks and the computational challenges of using them for data analysis when network weights are statistically modelled as being time-varying. More recent works also combined Kalman Filter [28,29] and state-space model [30] with deep learning, and can be interpreted as a subclass of HMNN, as they do not consider mixtures of Gaussians.

### 3.2. Bayesian DropConnect and DropOut

DropConnect [19,31] and DropOut [32,33] are well-known techniques to prevent NNs from overfitting. Ref. [34] proposes variational DropOut where the authors combined fully factorized Gaussian variational approximation with the local reparameterization trick to re-interpret DropOut with continuous noise as a variational method. Ref. [35] extensively treat the connections between DropOut and Gaussian processes, and they show how to train NNs with DropOut (or DropConnect [31]) in a variational Bayes setting. Our version of variational DropConnect has several common aspects with the cited works, but the novelty resides in the regularization being induced by the variational approximation’s choice and the corresponding reformulation of the reparameterization trick.

### 3.3. Bayesian Filtering

There are multiple examples of NN training through Bayesian filtering [36,37,38,39]. In particular, the recent work of [40] proposed AdaBayes and AdaBayes-SS, where updates resembling the Kalman filter are employed to model the conditional posterior distribution over a weight of an NN given the states of all the other weights. However, the main difference with HMNN is the dynamical evolution of the underlined NN, and indeed, Bayesian filtering methods, in this context, do not consider any change in time. Still from a Bayesian perspective, HMNN could also be used as time-evolving prior and speed-up training of static models [41].

### 3.4. Continual Learning

There are significant similarities between our work and continual learning methods. Here, we perform a quick overview of the most popular ones and we refer to [8] for a complete review. Elastic Weight Consolidation (EWC) [10] uses an L2-regularization that guarantees the weights of the NN for the new task being in the proximity of the ones from the old task. Variational continual learning (VCL) [11] learns a posterior distribution over the weights of an NN by sequentially approximating the true posterior distribution through variational Bayes and by propagating forward the previous variational approximation (this is like setting our transition kernel to a Dirac delta). Online Laplace approximation [12,42,43] proposes a recursive update for the parameters of a Gaussian variational approximation which involves a Hessian of the newest negative log-likelihood. None of the cited techniques builds dynamic models, and even if this could be solved by storing the weights at each training step, there is no forgetting, meaning that EWC, VCL, and Online Laplace focus on overcoming catastrophic forgetting and they are not able to avoid outdated information. Lastly, the most similar procedure to HMNNs is the one proposed by [9], where the authors performed model adaption with Bayes forgetting through the application of a stochastic kernel. However, HMMs are not even cited and the form of the kernel and variational approximation are restricted to Gaussian distributions, not mixtures.

## 4. Experiments

In this section, we test the performance of HMNNs. Similar to [7], we focus our study on simple feed-forward neural networks, leaving the exploration of more complex architectures for future work. Notably, in our experiments, the computational cost per time step is comparable to that of Bayes by Backprop.

Firstly, we empirically demonstrate that variational DropConnect, i.e., using (Equation 12) as a variational approximation, yields better performance than Bayes by Backprop [7] on MNIST [44]. Secondly, we provide an experiment to show how HMNNs work and how they can quantify uncertainty in a simple time-evolving classification setting built from the “two-moons” dataset [45]. We then highlight the ability of HMNNs to retrieve the evolution of true parameters in a conceptual drift scenario [9]. Specifically, we demonstrate that employing more complex variational approximations, compared to those used in [9], does not affect the retrieval process. Following this, we explore a more complex conceptual drift framework constructed from the MNIST dataset, comparing HMNNs to continual learning baselines [9,10,11]. Finally, we show that HMNNs can also be applied to one-step-ahead forecasting in time-series. Specifically, we address a next-frame prediction task in the dynamic video texture of a waving flag [46,47,48]. Additional experimental details are provided in Appendix B.

The experiments were run on three different clusters: BlueCrystal Phase 4 (University of Bristol), Cirrus (one of the EPSRC Tier-2 National HPC Facilities), and The Cambridge Service for Data-Driven Discovery (CSD3) (University of Cambridge).

### 4.1. Variational DropConnect

The experiment aims to understand if using a Gaussian mixture as a variational approximation can help improve Bayes by Backprop. We trained on the MNIST dataset with the same setup from [7]. We considered a small architecture with the vectorized image as input, 2 hidden layers with 400 rectified linear units [49,50] and a softmax layer on 10 classes as output. We considered a fully Gaussian HMNN, as in Section 2.3, with T=1, α=0 and μ=0 (kernel parameter), with 0 being the zero vector. Note that such an HMNN coincides with a single Bayesian neural network, meaning that we are simply training with Bayes by Backprop with the addition of variational DropConnect; see Appendix B for more details. We trained on about 50 combinations of the parameters $\left(\gamma^{v},\phi ,\sigma ,c\right)$ and learning rate, which were randomly extracted from pre-specified grids.

Model selection was performed according to a validation score on a held-out validation set. We cluster the 50 combinations of the parameters by their values of $\gamma^{v}$ and we report in Figure 2 the performance of the three best models per each $\gamma^{v}$ value on the held-out validation set.

We then selected the best models per each $\gamma^{v}$ and we computed the performance on a held-out test set; see Table 2. We found that values of $\gamma^{v}<1$ led to better performance, motivating the use of variational DropConnect as a regularization technique.

### 4.2. Illustration: Two Moons Dataset

In this subsection, we provide an illustration of the HMNN on the “two moons” dataset from “scikit-learn” [45]. This synthetic dataset produces two half circles in the plane, which are binary-classified. To create a time dimension, we sequentially rotated these half circles. Specifically, over t=0,1,2,3,4, we generated new data from the “two moons” dataset and then applied a rotation with an increasing angle by keeping the label as simulated. We also considered two scenarios: one where the “two moons” are well separated, and another where the “two moons” are overlapping; see Figure 3.

Given the two training sets, a fully Gaussian HMNN was run for a fixed set of hyperparameters; see Appendix B. For the neural network architecture, we considered a bidimensional vector as input, i.e., (x,y) location, 2 hidden layers with 50 rectified linear units, and a softmax layer on 2 classes as output. The result was an evolving in-time Bayesian neural network that is able to associate to each location on the plane a probability of being in one class or the other, and also to quantify uncertainty on these probabilities. Note that it is enough to report one of the two probabilities as the other one is the complement.

We illustrate these results in Figure 4, where a 95% credible interval was obtained via Monte Carlo sampling on the weights of the HMNN; see Section 2.4. Here, we can observe that the scenario with overlapping moon shows a smoother transition between the two classification regions, meaning that we do not know which label to guess. In terms of uncertainty in the overlapping example, we are generally less confident about our probability estimates compared to the well-separated one. Moreover, in the well-separated example, there are some blank regions between two moons where the HMNN is completely uncertain about what to guess.

This can be further checked by plotting the length of the credible interval; see Figure 5. We can observe that where the two classes overlap, the HMNN becomes less certain (bigger length of the credible interval) but it becomes completely uncertain (length close to one) in the region between the two moons and where no data are observed. Intuitively, with the yellow region in Figure 5, the HMNN expresses the following: “I do not know”. We remark that this is one of the most important features of the Bayesian approach, which allows the model to be uncertain about the prediction and warn the user beforehand.

### 4.3. Concept Drift: Logistic Regression

In this subsection, we compare the HMNN and the adaption with the Ornstein–Uhlenbeck process proposed by [9] on concept drift. As in [9], we consider a 2-dimensional logistic regression problem where the weights evolve in time: $w_{t}^{\left(1\right)}=10\sin\left(at\right)$ and $w_{t}^{\left(2\right)}=10\cos\left(at\right)$, where a=5 deg/sec and t=1,…,700. Precisely, per each time step, the data are generated in the following manner:(16)$$x_{t}^{\left(i\right)}∼U\left(\cdot |-3,3\right),\;y_{t}^{\left(i\right)}∼\mathrm{Be}\left(\cdot |\mathrm{sigmoid}\left(x_{t}^{\left(i\right)},w_{t}\right)\right),$$
where U(·|a,b) is the uniform distribution over the interval [a,b], i=1,…, 10,000 with 10,000 being the batch size per time step, $w_{t}=\left(w_{t}^{\left(1\right)},w_{t}^{\left(2\right)}\right)$ and $\mathrm{sigmoid}\left(x_{t}^{\left(i\right)},w_{t}\right)$ being the sigmoid function with weights $w_{t}$ and input $x_{t}^{\left(i\right)}$.

Given the training set, a fully Gaussian HMNN is run under multiple combinations of hyperparameters. The experiments showed a recovery of the oscillating nature of the weights in each combination. Figure 6 reports the results of one of the considered hyperparameters’ combinations and the findings from [9]. We can conclude that the HMNN is able, as for [9], to recover the sinusoidal curve of the true parameters even if the form of the posterior distribution approximation in the HMNN is chosen to be a mixture of Gaussians. Recall that the aim of this section was not classification accuracy but rather the recovery of the oscillating nature of the weights.

### 4.4. Concept Drift: Evolving Classifier on MNIST

In this subsection, we compare HMNNs with continual learning baselines when the data-generating distribution is dynamic. We decide to artificially generate such a dataset from MNIST with the following procedure:We define two labellers: $C_{1}$, naming each digit with its label in MNIST; $C_{2}$, labelling each digit with its MNIST’s label shifted by one unit, i.e., 0 is classified as 1, 1 is classified as 2, …, 9 is classified as 0.We consider 19 time steps where each time step *t* is associated with a probability $f_{t}\in \left[0,1\right]$ and a portion of the MNIST’s dataset $D_{t}$.At each time step *t*, we randomly label each digit in $D_{t}$ with either $C_{1}$ or $C_{2}$ according to the probabilities $f_{t},1-f_{t}$.

The resulting ${\left(D_{t}\right)}_{t=1,\ldots ,19}$ is a collection of images where the labels evolve in *t* by switching randomly from $C_{1}$ to $C_{2}$ and vice versa. Validation and test sets are built similarly. In such a scenario, we would ideally want to be able to predict the correct labels by sequentially learning a classifier that is capable of inferring part of the information from the previous time step and forgetting the outdated one. Note that when $f_{t}=0.5$, the best we can perform is a classification accuracy of 0.5 because $C_{1}$ and $C_{2}$ are indistinguishable.

We consider a fully Gaussian HMNN with $\mu =m_{t-1}$ (kernel parameter) to encourage a strong memory of the past. The evolving in-time NN is composed of the vectorized image as input, 2 hidden layers with 100 rectified linear units and a softmax layer on 10 classes as output. The parameters $\alpha ,\gamma^{v}$ are selected through the validation set while the other parameters are fixed before training. Along with the previous HMNN, we sequentially trained five additional models for comparison: variational continual learning (VCL), without coreset; Elastic Weight Consolidation (EWC), with tuning parameter chosen with the validation set; Bayes by Backprop trained sequentially on the dataset; Bayes by Backprop on the full dataset; and the adaption of Kurle et al.’s work with the Ornstein–Uhlenbeck process [9]. Selected graphical performances on the validation sets are displayed in Figure 7.

To test the method, we report the mean over time of the classification accuracy, which can be found in Table 3. For the HMNN, Kurle et al., Bayes by Backprop, EWC and VCL, we choose the parameters that perform the best on validation. We find that HMNN, the model by [9] and a sequential training of Bayes by Backprop perform the best. It is not surprising that continual learning methods fall behind. Indeed, EWC and VCL are built to preserve knowledge on the previous tasks, which might mix up $C_{1}$ and $C_{2}$ and confuse the network.

### 4.5. One-Step-Ahead Prediction for Flag Waving

We conclude by testing HMNNs on predicting the next frame in a video. The dataset is a sequence of images extracted from a video of a waving flag [48,51]. The idea is to create an HMNN where the neural network at time *t* can predict the next frame, i.e., the NN maps frame *t* in frame t+1. To measure the performance, we use the metric suggested in [51], which is a standardized version of the RMSE on a chosen test trajectory: (17)$$M\left(y_{1:T},{\hat{y}}_{1:T}\right):=\sqrt{\frac{\sum_{t=1}^{T}{∥y_{t}-{\hat{y}}_{t}∥}_{2}^{2}}{\sum_{t=1}^{T}{∥y_{t}∥}_{2}^{2}}},$$
where $y_{1:T}$ is the ground truth on frames 1,…,T, and ${\hat{y}}_{1:T}$ are the predicted frames. Unless specified differently, ${\hat{y}}_{1:T}$ is a sequence of one-step-ahead predictions. To have a proper learning procedure, we need multiple frames per time step, otherwise, the neural network would just learn the current frame. To overcome this problem, we created a sliding window with 36 frames, meaning that at time step *t*, we train on predicting frames t−35,…,t from frames t−36,…,t−1, with t>36. The choice of the length for the sliding window is empirical, as we tried multiple lengths and stopped at the first one that did not overfit the data inside the window (the same procedure was repeated for all the baselines).

Per each time step, we used a simple architecture of three layers with 500, 20, 500 rectified linear units, the vectorized previous frame as input and the vectorized current frame as output (the dimension was reduced with PCA). We considered a fully Gaussian HMNN, with $\mu =m_{t-1}$ (kernel parameter) and all the other parameters selected through random grid search and validation. Figure 8 compares HMNN predictions with different baselines: Bayes by Backprop trained sequentially on the sliding windows (column named BBP), DropConnect trained sequentially on the sliding windows (column named DropConnect), LSTM trained with the same sliding window size (column named LSTM), and a trivial predictor that uses the previous frame as forecasting for the current frame (column name: frame t−1). We noticed that the LSTM is prone to overfitting on the sliding window and subsequently predicting frame *t* using the last frame seen without any uncertainty. Similar issues appear in sequential DropConnect. This unwanted behaviour is probably due to the absence of uncertainty quantification, resulting in overconfidence in the considered predictions. HMNN and sequential BBP are less certain about prediction and they create blurred regions where they expect the image to change. This phenomenon is particularly evident in the last row of Figure 8. Table 4 summarizes the performances using metric (Equation 17). Overall, the HMNN performs better than the baselines and it is directly followed by the sequential BBP.

## 5. Discussion

We propose a hybrid model between the Bayesian neural network and FHMM called the Hidden Markov Neural Network (HMNN), where the posterior distribution over the weights is estimated sequentially through variational Bayes with an evolving prior distribution obtained by propagating forward the variational approximation from the previous time step through a stochastic transition kernel in the same vein as the assumed density filter [17,18]. We also propose a new variational approximation that induces a regularization technique called variational DropConnect, which resembles DropConnect [19] and variational DropOut [34], and we reformulate the reparameterization trick according to it. We test variational DropConnect on MNIST. We analyse the behaviour of HMNNs in a simple conceptual drift framework. We compare the HMNN with multiple baselines in a complex conceptual drift scenario and in time-series prediction. In all the experiments, our method compares favourably against the considered baselines, while allowing for uncertainty quantification. Compared to other techniques where deep learning and HMM are combined [28,29,30,43], HMNN uses a scale mixture of Gaussians as both variational approximations and transition kernel, allowing for a richer representation of both the dynamics of the weights and the filtering distribution.

The idea behind the HMNN is simple and it opens multiple research questions that can be answered in future works. Firstly, the quality of the approximation is not treated and we are also unsure about the rate of accumulation of the error over time. Theoretical studies on variational approximations could be used to answer this matter [52,53]. Secondly, we do not propose any smoothing algorithm, which could improve the performance of the HMNN and could open avenues for an EM estimation of the hyperparameters. These could also be estimated via Recursive Maximum Likelihood Estimation [30] or by approximating the likelihood via sampling from the variational approximation. Thirdly, we have limited our studies to neural networks with simple architecture; it is surely possible to use similar techniques in recurrent neural networks and convolutional neural networks to tackle more complicated applications. Finally, the variational DropConnect can be applied in multiple variational Bayes scenarios to check if it leads to better performances.

## Figures and Tables

**Figure 1 entropy-27-00168-f001:**
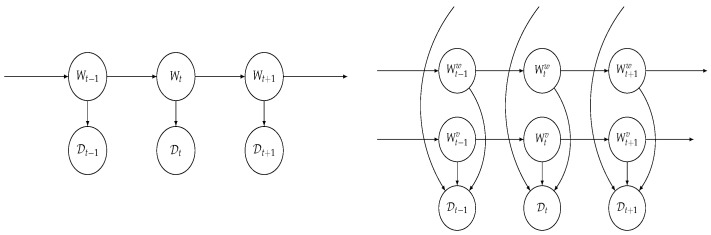
On the left: The conditional independence structure of an HMM. On the right: The conditional independence structure of an FHMM.

**Figure 2 entropy-27-00168-f002:**
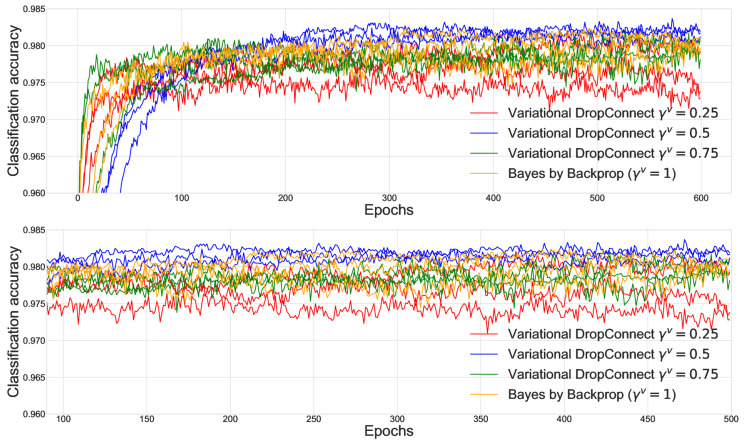
Performance on a validation set of a Bayes by Backprop with and without variational DropConnect. The plot on the bottom is a zoom-in of the plot on the top.

**Figure 3 entropy-27-00168-f003:**
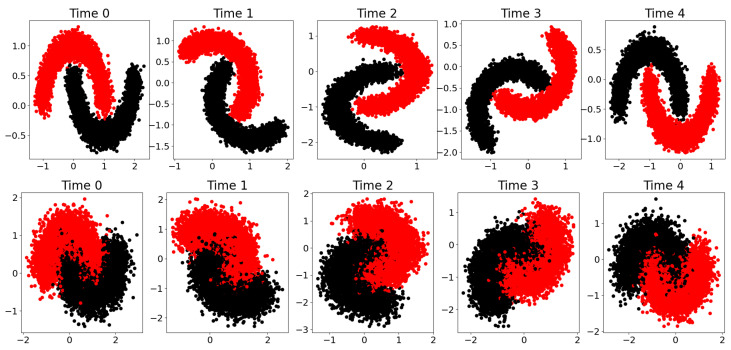
First row, well-separated “two moons”. Second row, overlapping “two moons”. Different columns are associated with different time steps. Different colours are associated with different labels.

**Figure 4 entropy-27-00168-f004:**
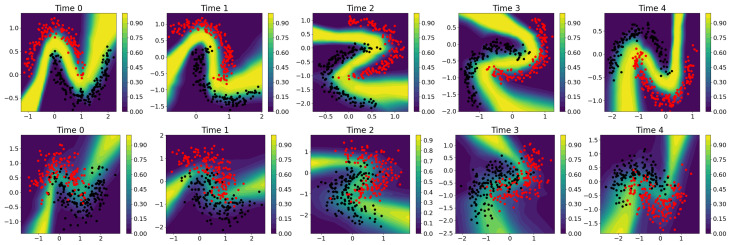
First row, well-separated “two moons”. Second row, overlapping “two moons”. Different columns are associated with different time steps. The plot shows the length of the 95% credible interval. The blue and yellow surface is the probability of prediction on the second class. Different coloured dots are associated with different labels.

**Figure 5 entropy-27-00168-f005:**
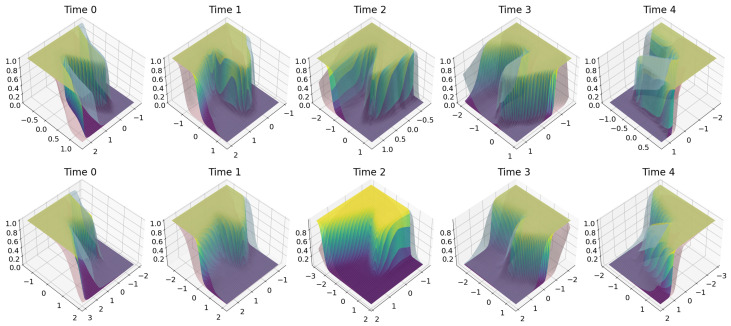
First row, well-separated “two moons”. Second row, overlapping “two moons”. Different columns are associated with different time steps. The blue and yellow surface is the probability prediction on the second class. Pink- and grey-shaded surfaces represent the 95% credible intervals.

**Figure 6 entropy-27-00168-f006:**
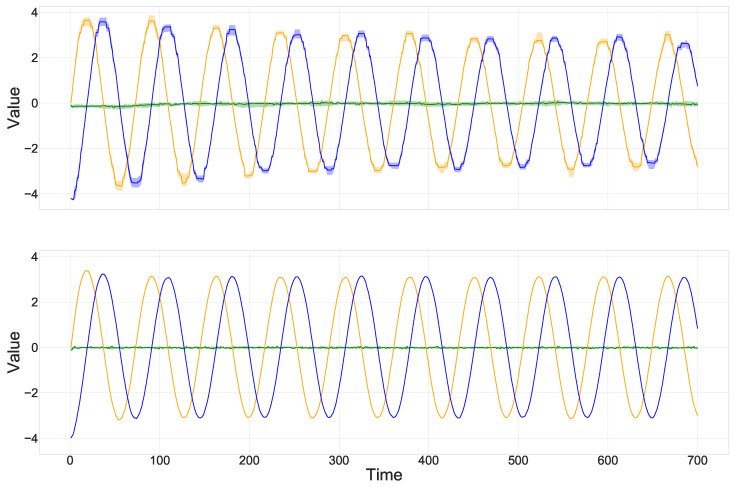
Mean of the approximate posterior distributions over 700 steps, where credible intervals were built from multiple runs. Orange stands for $w_{t}^{\left(1\right)}$, blue stands for $w_{t}^{\left(2\right)}$ and green is used for the bias. On the top, the HMNN. On the bottom, adaption with the Ornstein–Uhlenbeck process from [9].

**Figure 7 entropy-27-00168-f007:**
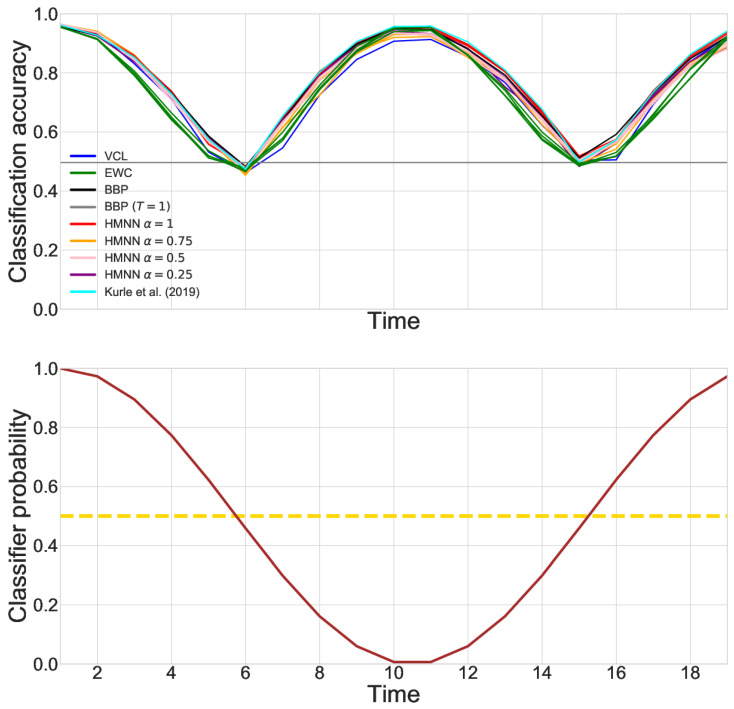
On the top, performances on a held-out validation set over time of the evolving classifiers obtained with different algorithms [9]. BBP refers to Bayes by Backprop trained sequentially. BBP (T = 1) refers to the training of Bayes by Backprop on the whole dataset. On the bottom, in brown, the evolution in time of the probability $f_{t}$ of choosing the labeller $C_{1}$, and in yellow, the value 0.5.

**Figure 8 entropy-27-00168-f008:**
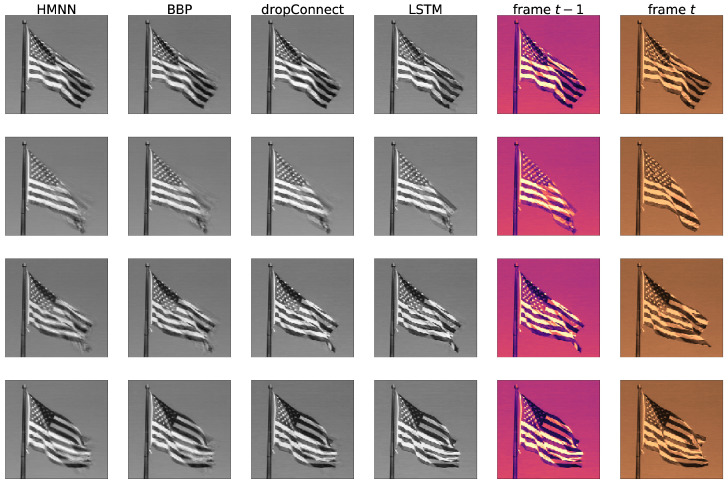
Columns show the prediction for different algorithms, with the last two being the last frame seen and the target frame. Rows display different time steps.

**Table 1 entropy-27-00168-t001:** Notation summary.

Notation	Meaning
$W_{t},D_{t}$	hidden weights of an NN and observed data at time *t*
$p(W_{t-1},W_{t})$	Markov transition kernel of the weights of the NN
$g(W_{t},D_{t})$	probability distribution of the data given the weights
*v*	a weight of the NN, also used to represent marginal quantities
$\pi_{t},{\tilde{\pi }}_{t}$	filtering distribution and its approximation at time *t*
$P,C_{t}$	prediction operator and correction operator at time *t*
$V_{Q}$	operator that minimizes the KL-divergence on the class Q
$q_{\theta },\theta$	variational approximation and its parameters
h(θ,·)	transformation function in the reparameterization trick
$\gamma^{v}$	scale mixture of Gaussian probability of weight *v*
$m_{t}^{v}$	scale mixture of Gaussian mean of weight *v* at time *t*
$s_{t}^{v}$	scale mixture of Gaussian standard dev. of weight *v* at time *t*
ϕ	scale mixture of Gaussian probability of transition kernel
μ	scale mixture of Gaussian stationary mean of transition kernel
α	scale mixture of Gaussian mean scaling of transition kernel
σ	scale mixture of Gaussian big-jump standard dev. of transition kernel
σ/c	scale mixture of Gaussians small-jump standard dev. of transition kernel

**Table 2 entropy-27-00168-t002:** Classification accuracy (the bigger the better) of best runs for MNIST on a held-out test set (MNIST test set). $\gamma^{v}=1$ refers to the case of Bayes by Backprop without variational DropConnect.

Parameter Value	Accuracy
$\gamma^{v}=0.25$	0.9838
$\gamma^{v}=0.5$	0.9827
$\gamma^{v}=0.75$	0.9825
$\gamma^{v}=1$ [7]	0.9814

**Table 3 entropy-27-00168-t003:** Classification accuracy on a held-out test set for the evolving classifier (the bigger the better). BBP refers to Bayes by Backprop trained sequentially. BBP (T = 1) refers to the training of Bayes by Backprop on the whole dataset.

Model	Accuracy
BBP (T = 1)	0.503
VCL	0.744
EWC	0.760
BBP	0.780
Kurle et al. [9]	0.784
HMNN	0.786

**Table 4 entropy-27-00168-t004:** Metric M value on the test set (the smaller the better).

Model	M
Trivial Predictor	0.2162
LSTM	0.2080
DropConnect	0.2063
BBP	0.1932
HMNN	0.1891

## Data Availability

A tutorial on the experiments and how to use the code is available at the GitHub repository: https://github.com/LorenzoRimella/HiddenMarkovNeuralNetwork (accessed on 20 December 2024).

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
