# Peer review of "Hidden Markov Neural Networks"

_entropy, 2025, doi:10.3390/e27020168_

Round 1

Reviewer 1 Report

Comments and Suggestions for Authors

This paper defines an evolving in-time Bayesian neural network. It is enriched with a stronger regularization technique called variational DropConnect. And the experiment which verified the effectiveness of the proposed method, is tested on MNIST. Thus, I recommend accepting the paper after major revision. I have put forward some suggestions for you to improve the paper:

1In the abstract, please emphasize the importance and significance of this research work.

2In Figure 2, there are three curves of the model under each parameter, which can further explain the three models.

3In Section 4, additional data sets be incorporated into each sub-section for the purpose of verification.

4Some preliminary knowledge can be added to help people understand the paper.

5In Section 2, there are many different symbols used, the meaning of these symbols can be declared in a table.

6In Figure 3, there is a lack of explanation for the classification accuracy of a classifier that draws multiple curves.

Author Response

We thank the reviewer for the insightful comments. In our report whenever referring to equations and lines we consider the revised manuscript, this should help track the changes.

Comment 1:

In the abstract, please emphasize the importance and significance of this research work.

Response 1:

We agree with the reviewer that the abstract did not effectively communicate the significance of our work. We have rewritten the abstract and emphasized that our hidden Markov Neural Network tries to strike a balance between adapting to new data and forgetting outdated information while also allowing for uncertainty quantification.

Comment 2:

In Figure 2, there are three curves of the model under each parameter, which can further explain the three models.

Response 2:

We thank the reviewer for the comment. The three best models were included for visual purposes in Figure 2, at test time we then considered a single one. We rephrased lines 276-281 to make this clearer.

Comment 3:

In Section 4, additional data sets be incorporated into each sub-section for the purpose of verification.

Response 3:

In section 4, we run an additional experiment, subsection 4.2, on the two-moons dataset, which we make time-evolving by rotating it over time. HMNN can quickly learn how to separate the two ``moons'' and quantify uncertainty. Specifically, we considered two scenarios: one where the two ``moons'' are well separated over time, and another where they overlap over time. We showed that the size of our credible intervals increases when the two ``moons'' overlap, and when no data are observed between the two classes. 

Comment 4:

Some preliminary knowledge can be added to help people understand the paper.}

Response 4:

We have simplified section 2 by adding some extra explanation on the factorization (1), lines 64-68, and the operators (3),(4),(9), lines 76-78 and lines 86-90 and lines 110-115. We have also written a new subsection, subsection 2.4, which explains how to measure performance and quantify uncertainty.

Comment 5:

In Section 2, there are many different symbols used, the meaning of these symbols can be declared in a table.

Response 5:

We thank the reviewer for this suggestion, we have added Table 1 which summarizes the key symbols used in the paper.

Comment 6:

In Figure 3, there is a lack of explanation for the classification accuracy of a classifier that draws multiple curves.

Response 6:

We have added a remark at the end of subsection 4.3 to clarify the aim of the experiment, lines 327-329.

Reviewer 2 Report

Comments and Suggestions for Authors

Comments are in the attached file.

Author Response

We sincerely thank the reviewer for the useful comments. In our report whenever referring to equations and lines we consider the revised manuscript, this should help track the changes.

Comment 1:

lines 54-55, The authors assume that the weights evolve independently from each other, and from formula (13) we can see, that the coordinates of the weights are independent on each other too. Is the restriction shown in formula (1) necessary?

Response 1:

We thank the reviewer for pointing this out. Yes, the factorization from (1) is necessary as it allows us to perform the prediction step at a low cost and in closed form. We have added a remark after (1), lines 64-67, to ensure this is clear. We also mention the possibility of having some form of block structure. This will make everything more costly as we have to track correlations across weights, but it might be an interesting extension, lines 67-68. 

Comment 2:

lines 73-74, thus the formula (2) is applicable for only linear activation function. How filtering can be applied to non-linear activation functions, e.g. ’relu’, ’sigmoid’, etc.?

Response 2:

We can apply our method to non-linear activation functions like relu, sigmoid, etc. We have rewritten lines 86-90 to ensure this is clear.

Comment 3:

All abbreviations should be explained when used for the first time, e.g. ELBO in line 80.

Response 3:

Thanks. We have ensured all the abbreviations are specified.

Comment 4:

In formula (7) we have the object $V_Q$, whereas in (8) we see $V_Q \rho$. This should be corrected.}

Response 4:

(8) is the recursion that defines HMNN, while (9) is the definition of the operator $V_Q$. To define the operator $V_Q$ we require a general probability distribution $\rho$ as it represents the operation of minimizing the KL in variational inference. Specifically in (8) we are minimizing the KL using $\rho = C_t P \tilde{\pi}_{t-1}$. We have remarked on the use of the operator's notation at the beginning of subsection 2.1, lines 76-78, we have rephrased lines 110-112 to clarify this, and we have added Table 1 to make a clear summary of all the notation. 

Comment 5:

lines 133-134. Are the parameters $\pi, \alpha, \sigma, c, \mu$ fixed during the learning process?}

Response 5:

We thank the reviewer for the question. The hyperparameters of the transition kernel are chosen via model selection on a held-out validation set. This means that we consider multiple combinations of these parameters, each combination is considered fixed during training, and, after training, we choose the best combination on a validation set. We did some rephrasing in the experiments to make sure this was clear throughout. As this is a really good point we have also included in the discussion the possibility of learning them via a likelihood approximation and EM, lines 410-414.

Comment 6:

in formula (11) the hyperparameter $\gamma^v$ corresponds to DropConnect technique to avoid overfitting during the learning process. In line 125 we read ”..$w^v$ is distributed as (11) which is equivalent to consider...”. Please explain proximity between (11) and (12).}

Response 6:

We thank the reviewer for allowing us to clarify this. (12) is the formulation of (11) via the reparameterization trick, which allows us to compute the gradient via Monte Carlo sampling. We have clarified this in lines 146-151. 

Comment 7:

Table 1 shows that the regularisation parameter $\gamma^v$ has a weak effect on accuracy. If possible, please explain these results.}

Response 7:

We have clarified that the experiment from subsection 4.1 aims to show that $\gamma^v<1$ improves accuracy by rephrasing lines 280-281. We have also included a new experiment that clarifies the aim and the use of HMNN in subsection 4.2.

Reviewer 3 Report

Comments and Suggestions for Authors

The manuscript presented a hidden Markov neural network. It is not recommended for acceptance.

1.      The paper is almost identical to the authors’ another work “Dynamic Bayesian neural networks” published on arxiv in 2020. The link is: https://arxiv.org/pdf/2004.06963.

2.      There have been numerous developments in neural networks with Markov chain and Bayesian neural networks in the past few years. What is the novelty of this work?

3.      All cited references are in 2019 or older. As mentioned in (2), the latest developments should be considered and discussed.

4.      The explanation in the theoretical is vague and the logic in different parts is hard to follow.

5.      The symbols in different sections are inconsistent.

6.      In the experiment, how the model selection is conducted?

7.      It is suspected whether the accuracy is affected by the statistical variation and the actual performance.

8.      The experiment is unconvincing to demonstrate the outperformance of the presented hidden Markov neural network over other existing methods. 

Author Response

We thank the reviewer for the time spent on our manuscript. We hope that the revised paper will clarify their concerns. In our report whenever referring to equations and lines we consider the revised manuscript, this should help track the changes.

Comment 1:

The paper is almost identical to the authors’ another work “Dynamic Bayesian neural networks” published on arxiv in 2020. The link is: https://arxiv.org/pdf/2004.06963.

Response 1:

That is actually the same paper and the journal encouraged us to submit it. We have changed the title because we thought ``Dynamic Bayesian Neural Network'', the old title, was misleading. We are in the process of updating our preprint to ensure it is clear that they are the same thing.

Comment 2:

There have been numerous developments in neural networks with Markov chain and Bayesian neural networks in the past few years. What is the novelty of this work?

Response 2:

To the author's knowledge, HMNN is the first evolving in-time neural network that is expressed as an HMM and where inference is pursued via a filtering algorithm. In the revised manuscript we significantly work on clarifying our contribution by rewriting the abstract, and by being more precise throughout the paper, see later comments.

Comment 3:

All cited references are in 2019 or older. As mentioned in (2), the latest developments should be considered and discussed.

Response 3:

We thank the reviewer for pointing this out. We have added more recent references in section 3 to ensure the paper is more up-to-date.

Comment 4:

The explanation in the theoretical is vague and the logic in different parts is hard to follow.

Response 4:

 In section 2, we have added a comment after (1) to explain the factorization, lines 64-68, we have remarked on the use of the operator notation, lines 76-78, we have clarified that we can handle any activation function, lines 86-90, and the use of the operator $V_Q$, lines 110-112. We have also added a comment on Assumed Density Filter, lines 113-115. In the revised manuscript we also have a new section 2.4 that specifies how HMNN can quantify uncertainty.

Comment 5:

The symbols in different sections are inconsistent.

Response 5:

We checked all our symbols for inconsistencies and we did not find any, but we understand that the notation becomes quickly heavy. In the revised manuscript we have added Table 1 to summarize the notation.

Comment 6:

In the experiment, how the model selection is conducted?

Response 6:

We thank the reviewer for helping us in clarifying this. Model selection is performed via a held-out validation set. In the revised manuscript we ensure this is clear in all the experiments.

Comment 7:

It is suspected whether the accuracy is affected by the statistical variation and the actual performance.

Response 7:

The appeal of the HMNN is not necessarily predictive accuracy but rather achieving similar accuracies to competitors while allowing for uncertainty quantification. We have emphasized this point way more in the revised manuscript by modifying the abstract.

Comment 8:

The experiment is unconvincing to demonstrate the outperformance of the presented hidden Markov neural network over other existing methods.

Response 8:

We connect with the previous comment and highlight once more that the aim of HMNN is to provide a way to quantify uncertainty for these evolving in-time NN while not losing performance in terms of accuracy. We have provided a new experiment in subsection 4.2 which should make this clearer and we hope this will satisfy the reviewer.

Round 2

Reviewer 1 Report

Comments and Suggestions for Authors

The author has thoroughly reviewed the reviewer's opinions, made pertinent revisions accordingly, and consequently, has agreed to accept the manuscript.

Author Response

We thank the reviewer for helping us in improving the manuscript.

Reviewer 3 Report

Comments and Suggestions for Authors

The improvements in the revised version are insufficient. It is not suggested for publication.

Author Response

We thank the reviewer for the comment.